# Mechanism of Skyrmion Attraction in Chiral Magnets near the Ordering Temperatures

**DOI:** 10.3390/nano13050891

**Published:** 2023-02-27

**Authors:** Andrey O. Leonov, Ulrich K. Rößler

**Affiliations:** 1Leibniz-Institut für Festkörper- und Werkstoffforschung Dresden (IFW Dresden), Helmholtzstraße 20, D-01069 Dresden, Germany; 2Department of Chemistry, Faculty of Science, Hiroshima University, Kagamiyama, Higashi-Hiroshima 739-8526, Hiroshima, Japan; 3International Institute for Sustainability with Knotted Chiral Meta Matter, Kagamiyama, Higashi-Hiroshima 739-8511, Hiroshima, Japan

**Keywords:** skyrmion, precursor effects, chiral magnets, Dzyaloshinskii–Moriya interaction, A-phase, conical spiral, 75.30.Kz, 12.39.Dc, 75.70.-i

## Abstract

Isolated chiral skyrmions are investigated within the phenomenological Dzyaloshinskii model near the ordering temperatures of quasi-two-dimensional chiral magnets with $C_{nv}$ symmetry and three-dimensional cubic helimagnets. In the former case, isolated skyrmions (IS) perfectly blend into the homogeneously magnetized state. The interaction between these particle-like states, being repulsive in a broad low-temperature (LT) range, is found to switch into attraction at high temperatures (HT). This leads to a remarkable confinement effect: near the ordering temperature, skyrmions exist only as bound states. This is a consequence of the coupling between the magnitude and the angular part of the order parameter, which becomes pronounced at HT. The nascent conical state in bulk cubic helimagnets, on the contrary, is shown to shape skyrmion internal structure and to substantiate the attraction between them. Although the attracting skyrmion interaction in this case is explained by the reduction of the total pair energy due to the overlap of skyrmion shells, which are circular domain boundaries with the positive energy density formed with respect to the surrounding host phase, additional magnetization “ripples” at the skyrmion outskirt may lead to attraction also at larger length scales. The present work provides fundamental insights into the mechanism for complex mesophase formation near the ordering temperatures and constitutes a first step to explain the phenomenon of multifarious precursor effects in that temperature region.

## 1. Introduction

Magnetic skyrmions are particle-like topological excitations with complex non-coplanar spin structure [1,2,3]. Recently, they have been observed in conducting and insulating helimagnets [4,5,6,7,8,9] as well as in frustrated (centrosymmetric) materials [10,11,12,13] under an applied magnetic field. Small size, topological protection and the ease with which ISs can be manipulated by electric currents generated much interest in using them as information carriers in memory and logic devices [14,15,16].

The stabilization mechanism of chiral ISs relies on the Dzyaloshinskii–Moriya interaction (DMI) [17,18,19,20], which is generally expressed as the energy terms with the first-order derivatives of the magnetization M with respect to the spatial coordinates, the so called Lifshitz invariants (LI) [21]:(1)$$L_{i,j}^{\left(k\right)}=M_{i}\partial M_{j}/\partial x_{k}-M_{j}\partial M_{i}/\partial x_{k}.$$
These LIs arise in certain combinations depending on the crystal symmetry of an underlying chiral magnet [1]. Moreover, they lead to a unique internal structure of ISs by defining their vorticity and helicity. For cubic helimagnets belonging to 23 (T) (as MnSi [4], FeGe [5], and other B20 compounds) and 432 (O) crystallographic classes, ISs are truly three-dimensional (3D) states. Their two-dimensional (2D) pattern on the plane xy is known to be modified by the LI along *z* axis, which becomes evident, e.g., in additional surface twists near surfaces of thin-film samples [22,23,24]. Moreover, skyrmions can be embraced by the conical phase with the wave vector along *z*, which shapes IS internal structure and stipulates attractive nature of inter-skyrmion potentials [25,26]. In, e.g., the $C_{nv}$ symmetry class, however, no Lifshitz invariants are present along the high-symmetry *z* axis [1]. Thus, ISs retain their 2D structure in both bulk and nanofabricated magnets [27,28].

In frustrated magnets, the limitations of the Derrick-Hobart theorem are overcome by the competing ferromagnetic and antiferromagnetic exchange interactions between spins, e.g., on a triangular lattice, which generate higher-order derivative terms in the fundamental continuum form of the magnetic free energy density [10,11,12,13]. In these frustrated magnets, the spin direction in the skyrmion centre is still opposite to that of the applied magnetic field. However, the topological charge of skyrmions can be both +1 (skyrmions) and −1 (anti-skyrmions) owing to the arbitrary sign of vorticity. In addition, the helicity angle χ is arbitrary and represents a zero mode. χ=0,π corresponds to skyrmions carrying a monopole moment, $A\propto \sum_{j}x_{j}\cdot S_{j}$, whereas χ=±π/2 distinguishes skyrmions with a toroidal moment, $T^{z}\propto \sum_{j}{\left[x_{j}\times S_{j}\right]}^{z}$ [29]. These degrees of freedom, however, do not exist for skyrmions in chiral magnets. As a matter of fact, DMI selects only one type of IS according to the symmetry arguments, and other localized solitonic solutions describe metastable or instable particles with higher energy.

In contrast to the fixed and rigid non-collinear spin pattern of skyrmions in chiral magnets, the skyrmions in frustrated magnets, thus, have internal degrees of freedom. It is instructive to compare the properties, shape and mutual interactions of skyrmions in frustrated magnetics to those in chiral helimagnets with DMIs, regarding these internal degrees of freedom. Interestingly, the toroidal and monopole moment densities within these skyrmions exhibit periodic sign reversals along the radial directions with growing distance *r* from the IS center [11,12]. In these patterns, the spins at the skyrmion outskirt undergo fan oscillations with decaying amplitude, which also give rise to a number of minima in the skyrmion–skyrmion interaction potentials $U(r_{12})$. On the one hand, mutual skyrmion interaction depends on the topological charges of skyrmions. Two skyrmions or two antiskyrmions attract each other at distances of the order of the skyrmion diameter, the maximal reduction of energy being ∼10% of the skyrmion eigen-energy, which corresponds to the “deepest” minimum of $U(r_{12})$. Hence frustrated ISs tend to form clusters with a short-range crystal order, resembling the clustering of vortices in 1.5-type superconductors [30]. On the other hand, interactions between frustrated skyrmions depend on their helicities. This effect couples helicity dynamics to the translational motion of skyrmions during, e.g., current-driven dynamics [11,12]. The oscillating interactions between skyrmions in the frustrated magnets are in sharp contrast with skyrmion–skyrmion interaction in non-centrosymmetric systems, which do not show fan-like oscillations and repel each other at all distances [31]. Hence, one would not expect chiral helimagnets to display similar attractive and complex skyrmion–skyrmion interactions as in frustrated systems at all.

In the present paper, chiral isolated skyrmions are analysed close to the ordering temperature using the phenomenological Dzyaloshinskii model supplemented with the basic Landau expansion for the homogeneous part of the free-energy. This study is motivated by the well-known fact that the longitudinal magnitude of the order-parameter, here the ferromagnetic magnetization *M*, and the non-collinear twisting are not independent. The non-collinear twisted pattern of skyrmions, or any other localized spin-pattern like kinks, becomes inhomogeneous also with respect to the local magnetization *M* near the ordering temperature, where the order parameter is soft both in transversal and in longitudinal direction. The inherent frustration by the LIs therefore does not only twist the orientable order-parameter considered as a field distribution of fixed length vector, but it also modulates its magnitude. In a short manner, we may say that for the order-parameter fields in Dzyaloshinskii models the Goldstone modes, i.e., transversal spin distortions, and the longitudinal Higgs mode are intertwined because of the gauge-background encoded in Equation (Equation 1). This behavior leads to interesting consequences. The inhomogeneous nature of periodic skyrmion lattice solutions has been described earlier [32,33]. Whereas in a broad temperature range the magnetization modulus |M| remains constant and is independent on the value of the applied magnetic field, its longitudinal stiffness decreases near the ordering temperature, and spatial longitudinal modulations of the magnetization become a sizable effect. We show that isolated skyrmions develop haloes of damped and oscillatory spin twistings in radial directions around their rigid core, accompanied by oscillations of *M*. This modification of the skyrmion pattern creates attractive inter-skyrmion pair-potentials, and achieves also a longer range of the interaction. The crossover from the repulsive to attractive potential is a consequence of the coupling between the magnitude and the angular part of the magnetization M and occurs at the so-called confinement temperature.

First, from numerical investigations on 2D models of isotropic chiral ferromagnets, we investigate the properties of chiral ISs and find complex combinations of rotational and longitudinal degrees of freedom. In particular, we focus on the internal structure of ISs undergoing a collapse process at high fields. Second, we extend the paradigm to 3D-ISs surrounded by the conical phase in bulk cubic helimagnets and confirm the same attracting interaction due to the oscillatory asymptotic behaviour. Our results identify a fundamental mechanism in chiral magnets and similar systems described by Dzyaloshinskii models, in that particle-like solutions display complex oscillatory transformations of their patterns near phase transition, as their orientational and longitudinal ordering modes are coupled. For chiral helimagnets this point to the long-range entangled character of the skyrmion–skyrmion interaction which underlies precursor phenomena in chiral magnets near the ordering temperatures, as observed in experiment [34,35,36].

## 2. Phenomenological Theory

Within the phenomenological theory introduced by Dzyaloshinskii [17,18,19,37] the magnetic energy density of a non-centrosymmetric ferromagnet with spatially dependent magnetization M can be written as
(2)$$W_{0}\left(M\right)=A\sum_{i,j}{\left(\frac{\partial m_{j}}{\partial x_{i}}\right)}^{2}+D\;w_{D}\left(M\right)-M\cdot H$$
where m is the unit magnetization vector; A>0 and *D* are coefficients of exchange and Dzyaloshinskii–Moriya interactions; H is a magnetic field applied along the *z* axis; $x_{i}$ are the Cartesian components of the spatial variable. The functional (Equation 2) contains only the basic interactions essential to stabilize skyrmionic states in noncentrosymmetric ferromagnets. In this paper, we omit some less important energy contributions such as magnetic anisotropy, stray-field energy, magneto-elastic coupling etc.

First, we restrict all calculations to the two-dimensional case on the xy plane. As an instructive example, we consider the DMI energy density inherent to magnets with the $C_{nv}$ symmetry [38]:(3)$$w_{D}=m_{x}\partial_{x}m_{z}-m_{z}\partial_{x}m_{x}+m_{y}\partial_{y}m_{z}-m_{z}\partial_{y}m_{y},$$
where $\partial_{x}=\partial /\partial x,\;\partial_{y}=\partial /\partial y$.

Near the ordering temperatures the magnetization amplitude varies under the influence of the applied magnetic field and temperature. Commonly this process is described by supplementing the magnetic energy (Equation 2) with the additional homogeneous free energy term f(M) [37].
(4)$$f\left(M\right)=a_{1}M^{2}+a_{2}M^{4}$$
where $a_{1}$ and $a_{2}$ are corresponding coefficients of Landau expansion. The magnetic moment M enters only in even powers due to the time reversal symmetry. The case $a_{1}=0$ corresponds to the critical temperature $T_{c}$ where spontaneous magnetization appears. Therefore, one can write $a_{1}$ in the form:(5)$$a_{1}=J\left(T-T_{c}\right).$$

By rescaling the spatial variable, the magnetic field, and the magnetization
(6)$$\begin{array}{c} x=\frac{x}{L_{D}},\;h=\frac{H}{H_{0}},\;m=\frac{M}{M_{0}} \end{array}$$
where
(7)$$L_{D}=\frac{A}{D},\;H_{0}=\kappa M_{0},\;M_{0}=\sqrt{\frac{\kappa }{a_{2}}},\;\kappa =\frac{D^{2}}{A},$$
energy density can be written in the following reduced form
(8)$$\begin{array}{c} \Phi ={\left(\mathrm{grad}\;m\right)}^{2}-w_{D}\left(m\right)-h\cdot m+am^{2}+m^{4}. \end{array}$$
Coefficient *a* is expressed as
(9)$$a=\frac{a_{1}}{\kappa }=\frac{J(T-T_{c})}{\kappa }.$$
Alongside with three internal variables (components of the magnetization vector m) functional (Equation 8) includes only two control parameters, the reduced magnetic field amplitude, *h*, and the “effective” temperature a(T) (Equation 9). By direct minimization of Equation (Equation 8) one can derive one-dimensional (spirals) and two-dimensional (skyrmions) phases with the propagation directions perpendicular to the polar axis, as dictated by the form of DMI (Equation 3). In the present manuscript, however, we analyze only solutions for localized isolated skyrmions (mentioned 1D and 2D bound states will be considered elsewhere).

## 3. Solutions for High-Temperature Isolated Skyrmions

### 3.1. Equations

Isolated skyrmions can be thought of as isolated static solitonic textures localized in two spatial directions. The magnetization in the center of skyrmion pointing opposite to an applied magnetic field rotates smoothly in all radial directions and reaches the orientation along the field at the outskirt of skyrmion. We will use the spherical coordinates for the magnetization in ISs,
(10)m=m(ρ)(sinθ(ρ)cosψ(φ),sinθ(ρ)sinψ(φ),cosθ(ρ))
and cylindrical coordinates for the spatial variables [27,28], r=(ρcosφ,ρsinφ). Thus, the structure of isolated skyrmions near the ordering temperature is characterized by the dependence of the polar angle θ(ρ) and modulus m(ρ) on the radial coordinate ρ with ψ=φ. The total energy *E* of such a skyrmion after substituting ψ(ϕ) (see dependences ψ(ϕ) corresponding to different crystallographic classes in Ref. [1]) is as follows:(11)$$E=2\pi \int_{0}^{\infty }\Phi \left(m,\theta \right)\rho d\rho$$
where energy density is
(12)$$\begin{array}{cc} \Phi =\; & m_{\rho }^{2}+m^{2}\left[\theta_{\rho }^{2}+\frac{\sin^{2}\theta }{\rho^{2}}-\theta_{\rho }-\frac{\sin\theta \cos\theta }{\rho }\right]+\\ \; & am^{2}+m^{4}-hm\cos\theta . \end{array}$$

The Euler equations for the functional (Equation 8)
(13)$$\begin{array}{cc} \; & m^{2}\left[\theta_{\rho \rho }+\frac{\theta_{\rho }}{\rho }+\frac{\sin\theta \cos\theta }{\rho^{2}}+\frac{2\sin^{2}\theta }{\rho }-h\sin\left(\theta \right)\right]+\\ \; & 2\left(\theta_{\rho }-1\right)m_{\rho }=0,\\ \; & m_{\rho \rho }+\frac{m_{\rho }}{\rho }+m\left[\theta_{\rho }^{2}+\frac{\sin^{2}\theta }{\rho^{2}}+\theta_{\rho }+\frac{\sin\theta \cos\theta }{\rho }\right]+\\ \; & 2am+4m^{3}-h\cos\left(\theta \right)=0 \end{array}$$
with boundary conditions
(14)$$\theta \left(0\right)=\pi ,\theta \left(\infty \right)=0,m\left(\infty \right)=m_{0},m\left(0\right)=m_{1}$$
describe the structure of isolated skyrmions. The magnetization of the homogeneous phase $m_{0}$ is derived from equation:(15)$$2am_{0}+4m_{0}^{3}-h=0.$$

Equation (Equation 13) can be solved numerically. However, before to consider typical solutions θ(ρ), m(ρ) of Equation (Equation 13) we focus on the asymptotic behavior of skyrmion solutions and some remarkable results that can be drawn by simple means. For 1D kinks such an analysis was done in Refs. [39,40].

### 3.2. Asymptotic Behaviour of Skyrmion Solutions

The asymptotic behaviour of isolated skyrmions bears exponential character [27,28]:(16)$$\Delta m=\left(m-m_{0}\right)\propto \exp\left(-\alpha \rho \right),\;\theta \propto \exp\left(-\alpha \rho \right).$$
By substituting these asymptotes into the linearized Euler Equation (Equation 13) for ρ→∞
(17)$$\begin{array}{c} \Delta m_{\rho \rho }-m_{0}\theta_{\rho }-\frac{1}{2}f_{mm}\left(m_{0}\right)\Delta m=0,\\ m_{0}^{2}\theta_{\rho \rho }-\frac{h\theta }{2}m_{0}+m_{0}\Delta m_{\rho }=0 \end{array}$$
one finds three distinct regions in the magnetic phase diagram on the plane (a,h) with different character of skyrmion–skyrmion interactions (Figure 1): *repulsive* interactions between isolated skyrmions occur in a broad temperature range (area (I)) and is characterized by real values of parameter α∈ℜ, the magnetization in such skyrmions has always “right” rotation sense in accord with DMI; at higher temperatures (area (II)) the skyrmion–skyrmion interaction changes to *attractive* character with α∈C; finally, in area (III) near the ordering temperature, $a_{N}=0.25$, only strictly confined skyrmions exist with imaginary values of parameter α∈ℑ.

The equation for parameter α obtained from (Equation 17)
(18)$$\alpha^{4}+\alpha^{2}\left[-2\;a-8\;m_{0}^{2}+1\right]+\left(a+6\;m_{0}^{2}\right)\;\left(a+2\;m_{0}^{2}\right)=0$$
allows to write the equation for the line separating different regions:(19)$$\begin{array}{cc} \; & {\left(f_{mm}\left(m_{0}\right)-\frac{f_{m}\left(m_{0}\right)}{m_{0}}\right)}^{2}-\\ \; & 4\left(f_{mm}\left(m_{0}\right)+\frac{f_{m}\left(m_{0}\right)}{m_{0}}\right)+4=0,\\ \; & h=f_{m}\left(m_{0}\right). \end{array}$$
We note that this equation is written for a general term *f*. For the case of Landau expansion (Equation 4) the separating line looks like
(20)$$\begin{array}{c} h^{⋆}=\sqrt{2\pm P(a)}\left(a+1\pm P\left(a\right)/2\right),\;P\left(a\right)=\sqrt{3+4a},\; \end{array}$$
with turning points *p* ($-0.75,\sqrt{2}/4$), *q* ($0.06,0.032\sqrt{5}$), and *u* (−0.5, 0) (solid line in Figure 1). Here, we used first and second derivatives, $f_{m}\left(m_{0}\right)$ and $f_{mm}\left(m_{0}\right)$, calculated from Equation (Equation 4).

### 3.3. The Internal Structure of Confined Isolated Skyrmions

To obtain numerically rigorous solutions for isolated skyrmions, one has to solve the system of differential Equation (Equation 13) with the boundary conditions (Equation 14). We use the “pure” shooting method, where the integration proceeds from the initial point ρ=0 to a distant point ρ→∞, and we try to match boundary conditions (Equation 14) at the end of the integration. We use the same numerical routines as explicitly described in Ref. [41] (Chapter 17) and therefore omit their description here.

The typical solutions as profiles θ(ρ), m(ρ) for isolated skyrmions in each region are plotted in Figure 1b,c. Since in the region II the exponents α are complex numbers, the profiles display antiphase oscillations (Figure 1c). Rotation of the magnetization in such an isolated skyrmion contains two types of rotation sense: if rotation has “right” sense according to the DMI (blue shaded parts in Figure 1c), the modulus increases, and otherwise, modulus decreases in parts of the skyrmion with “wrong” rotation sense (red shaded parts). Such a unique rotational behaviour of the magnetization is a consequence of the strong coupling between two order parameters of Equation (Equation 13)—modulus *m* and angle θ.

The coupling of angular and longitudinal order parameters may be so strong, that oscillations in the asymptotics of isolated skyrmions do not diminish. The purely imaginary exponent α then reflects the region of strict confinement. For −0.5<a<0.25 line (Equation 20) delimits a small pocket (III) in the vicinity of the ordering temperature. Within this region skyrmions can exist only as bound states and drastically differ from those in the main part of the phase diagram.

## 4. The Properties of Confined Isolated Skyrmions

### 4.1. Collapse of Skyrmions at High Fields

The solutions of Equation (Equation 13) exist only below a critical line $h_{0}\left(T\right)$ (Figure 1a). As the applied field approaches this line, the magnetization in the skyrmion center $m_{1}$ (Equation (Equation 14)) gradually shrinks (Figure 2b), and as $m_{1}$ becomes zero, the skyrmion collapses. This is in contrast to low-temperature skyrmions which exist without collapse even at very large magnetic fields [27,28] and are protected by the stiffness of the magnetization modulus which maintains topological stability of skyrmions. At high temperatures the softness of the magnetization amplitude allows to destroy the core of the skyrmion by “forcing” through the magnetization vector $m_{1}$ along the applied field. The radial profile for the angle θ(ρ) nevertheless displays an increasing localization under fields, as it was also noted for LT-skyrmions (Figure 2a). As an example, we illustrated the magnetization process for an isolated skyrmion for a=−0.5 (Figure 2).

The topological protection of the LT-skyrmions, however, is strictly enforced only in the continuum, whereas the atomic lattice of real materials underlies their collapse. Recently, experimental identification of two distinct skyrmion collapse mechanisms, radial symmetric and chimera-type mechanisms, was reported in biatomic Pd/Fe bilayers with face-centred cubic (fcc) stacking on Ir(111) [42]. In this sense, the mechanism considered in the present manuscript is a high-temperature analogue of a radial symmetric skyrmion collapse in Ref. [42].

### 4.2. Inter-Skyrmion Attraction

For the purpose of investigation of skyrmion–skyrmion interaction, we introduce two skyrmions into a square sample and define the interaction energy per skyrmion $\varepsilon_{\mathrm{int}}$ versus the inter-skyrmion distance *L* (Figure 3). Due to the strongly oscillatory character of this dependence two isolated skyrmions will tend to locate at some discrete, equilibrium distances from each other and to be placed in minima of inter-skyrmion energy $\varepsilon_{\mathrm{int}}$ (Figure 3c). On the other hand, a single isolated skyrmion (minimum corresponding to L=0) cannot elongate into a pair of skyrmions because of the high potential barrier toward the minimum with finite *L* (the first deepest minimum of $\varepsilon_{\mathrm{int}}$). Adding skyrmions one by one an optimal number of skyrmions in the cluster is found - the isolated skyrmions tend to form a hexagonal lattice with the densest space filling. The deepest minimum of $\varepsilon_{\mathrm{int}}$ for two interacting isolated skyrmions is very close to the period of hexagonal skyrmion lattice existing for the same control parameters.

### 4.3. Phenomenon of Skyrmion Confinement

The confinement temperature $a_{L}=-0.75$ subdivides the temperature interval in low- and high-temperature parts: (i) in the main part ($a<a_{L}=-0.75$) the rotation of the local magnetization vector determines the chiral modulation, while the magnetization amplitude remains constant; (ii) at high temperatures ($a_{L}=-0.75<a<a_{N}=0.25$) spatial variation of the magnetization modulus becomes a sizeable effect, and strong interplay between longitudinal and angular variables is the main factor in the formation and peculiar behaviour of chiral modulations in this region.

The characteristic temperature $a_{L}$ is of fundamental importance for chiral magnets. It is of the same order of magnitude as the temperature interval
(21)$$\left(a_{N}-a_{C}\right)\propto \frac{D^{2}}{A}$$
(Figure 1a), where chiral couplings cause inhomogeneous precursor states around the magnetic order temperature (for details see Ref. [32]). Here, $a_{C}$ is the conventional Curie temperature for centrosymmetric systems. When the temperature drops below $a_{C}$, the energy density of a ferromagnetically spin-aligned state is the lowest one. Dzyaloshinskii–Moriya interactions with negative energy density favour the rotation of the moments. Therefore, the transition to the ferromagnetic state is preceded by a transition to modulated states at the temperature $a_{N}$. Due to the relativistic origin and corresponding weakness of the DM exchange the shift
(22)$$\Delta a_{1}=a_{N}-a_{C},$$
as well as
(23)$$\Delta a_{2}=a_{N}-a_{L},$$
are small. For MnSi $\Delta a_{1}$ is estimated to be 0.9 K [32]. The shift $\Delta a_{2}$ is estimated to be three times as large as $\Delta a_{1}$ [31,35]. The crossover and confinement effects arise as a generic property of the asymptotic behavior of chiral solitons at large distances from the core. These effects also apply to kinks [39,40] and hopfions [43].

## 5. Three-Dimensional Attracting Skyrmions within the Conical Phase of Cubic Helimagnets

For cubic helimagnets mentioned in the introduction Dzyaloshinskii–Moriya interactions are reduced to the following form:(24)$$w_{D}=\left(L_{yx}^{\left(z\right)}+L_{xz}^{\left(y\right)}+L_{zy}^{\left(x\right)}\right)=m\cdot \mathrm{rot}m.$$

In these material systems, isolated skyrmions retain their two-dimensional structure as long as they are surrounded by the homogeneous state for relatively high magnetic fields. In this case, the 2D structure in the plane xy is replicated along the third *z* direction. At the same time, at some moderate field values, the homogeneous state is replaced by the conical spiral with the propagation vector along the field. The structure of the cone is characterized by the constant polar angle of the magnetization, $\theta_{cone}=2h/m$, linearly varying azimuthal angle, ψ=z/2, and by the fixed value of the modulus $m_{cone}=\sqrt{(0.25-a)/2}$. For the chosen control parameter set, a=0.2 and h=0.03, the modulus and the $m_{z}$-component acquire the following values: $m_{cone}=0.1581$, $m_{z}=2h=0.06$, which are in agreement with the numerical results in Figure 4.

The internal spin pattern of IS with its axis along the wave vector of the conical phase is depicted in Figure 4. In the field range of cone stability, the distribution of the $m_{z}$-component in a cross-section xy splits into the central core region that nearly preserves the axial symmetry (Figure 4a) and the domain-wall region, connecting the core with the embedding conical state. This part of the skyrmion cross-section is *asymmetric* and accommodates a crescent-like shape, which undergoes additional screw-like modulation along *z* axis, matching the rotating magnetization of the conical phase. Remarkably, the distribution of the modulus *m* is more involved (Figure 4b): the modulus exhibits elongation in the crescent part and reduction on the opposite side from the skyrmion center.

A convenient way to depict these non-axisymmetric skyrmions, which has been proven to be particularly illustrative in addressing the character of skyrmion–skyrmion interaction [26], is as follows (Figure 4c): we extract the spins corresponding to the conical phase and then plot the remaining spins as spheres colored according to their magnitude. In this way, all intricate details of the internal structure are explicitly exposed, which makes skyrmion crosscuts along different directions superfluous. In particular, in Figure 4c we depict the 3D-model of the modulus *m*. From the point of view of the modulus, ISs are composed of a cylinder-like (blue) core centered around the magnetization opposite to the field, a thick (light-red) coil with the magnetization along the field, and somewhat “thinner” (dark-red) coil, which indicates the “ripples” of the magnetization considered before for 2D ISs. The 2D-cross-section along the skyrmion center in Figure 4d shows oscillating $m_{z}$-component and the modulus *m*. The oscillatory behavior of the magnetization manifests itself also in the color plot of the energy density averaged over the cone period along *z*. In Figure 4e, the energy pattern of a LT-skyrmion with the “positive” core in the center and the “negative” ring around it, is complemented by smaller rings with alternating energy density.

The internal structure of LT-ISs embraced by the conical phase has been extensively investigated in our previous papers [25,26]. In the LT-region, only one, the most pronounced coil, persists in 3D-models of the $m_{z}$-component, which originates from the domain boundary between an isolated skyrmion and the surrounding conical phase. The 3D model of the modulus (Figure 4c) becomes redundant since |m|=const. By theoretical modeling [26] and from the direct experimental observation [25], such skyrmions were shown to attract each other. Then, the cluster formation of such LT-ISs was envisioned as a process of zipping loops: a coil of one skyrmion penetrates the voids between the coils of another one. By this, the magnetization on the way from the center of one skyrmion to the center of another rotates as in ordinary axisymmetric skyrmions. We emphasize that the attracting interaction originates owing to the incompatibility of isolated skyrmions with the surrounding conical phase and from the necessity to adjust their internal structure at each elevation along *z* axis.

In the “soft” HT version of such isolated skyrmions, in addition to the global energy minimum of skyrmion–skyrmion interaction potential, there should be a number of smaller minima presumably formed owing to the oscillatory asymptotics, as was described before for 2D ISs. Thus, isolated skyrmions near the ordering temperatures become “aware” of each other’s presence via oscillatory magnetic ripples. They may start moving towards each other and gather into clusters thus occupying the deepest minimum of the mutual interaction potential.

## 6. Conclusions

Using the basic Dzyaloshinskii theory for isotropic chiral magnets, we thoroughly scrutinized the internal structure of isolated 2D skyrmions within the homogeneous state as well as of 3D skyrmions surrounded by the conical spiral in chiral magnets near the ordering temperature. In both cases, the magnetization exhibits coupling of longitudinal and orientational degrees of freedom, which becomes apparent in the fine-tuning of the skyrmion core structures and/or in the oscillatory “ripples” of the magnetization encircling the skyrmion. As a result, the skyrmionic particles develop attractive soliton–soliton interactions and ultimately become confined in extended clusters or textures. The theory and results from numerical simulations demonstrate why a multitude of different small pockets of different phases at the experimental phase diagrams is generically expected in a distinct temperature interval, interleaved between paramagnetic and helix magnetic state. Generically, the attractive couplings between the localized chiral patterns may yield first-order phase transition, between a paramagnetic-like skyrmionic precursor and the helical ground state, as reported for FeGe [34], similarly to the behavior of flux-line lattices in the type-1.5 superconducting systems. Furthermore, the zero-field fully chiral precursor state in MnSi [36] hints at the existence of an amorphous phase of interacting skyrmions, which corresponds to the fully confined region III of the basic phase diagram elucidated here. Hence, the confinement effects of chiral skyrmions strongly change the picture of the formation and evolution of chiral modulated textures and may shed new light on the problem of precursor states observed in chiral magnets.

## Figures and Tables

**Figure 1 nanomaterials-13-00891-f001:**
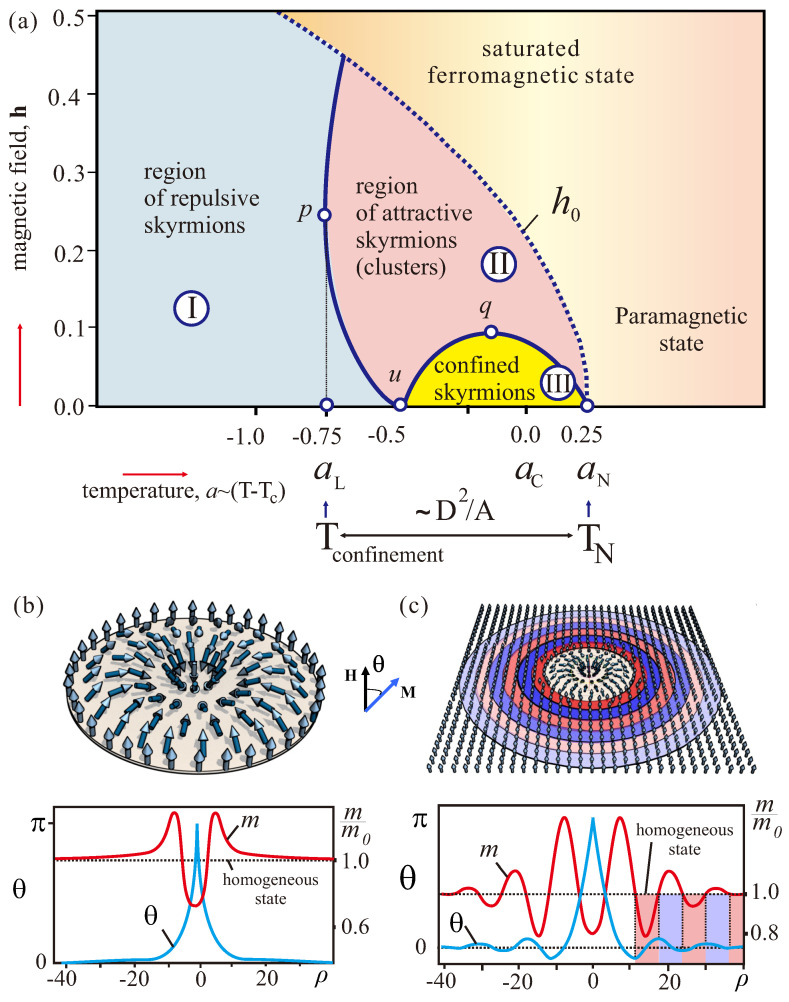
(color online) (**a**) The diagram on plane (a,h) showing the regions with different types of skyrmion–skyrmion interaction according to the analysis of IS asymptotic behaviour: I—repulsive interaction between isolated skyrmions; II—attractive inter-skyrmion interaction; III—the region of skyrmion confinement. Solid line is defined by Equation (Equation 20): the turning points have the following coordinates—*p* ($-0.75,\sqrt{2}/4$), *q* ($0.06,0.032\sqrt{5}$), and *u* (−0.5, 0). Above the dotted line $h_{0}$ no isolated skyrmions can exist, since they collapse. (**b**) Dependences of angular θ and longitudinal *m* order parameters on polar coordinate ρ for isolated skyrmion in region I (a=−1,h=0.4). (**c**) θ(ρ) and m(ρ) for isolated skyrmion in region II (a=0.21,h=0.05). Shaded parts of oscillating profiles emphasize “wrong” and “right” rotational senses according to DMI. The oscillations are damped and end at the level of the homogeneous state.

**Figure 2 nanomaterials-13-00891-f002:**
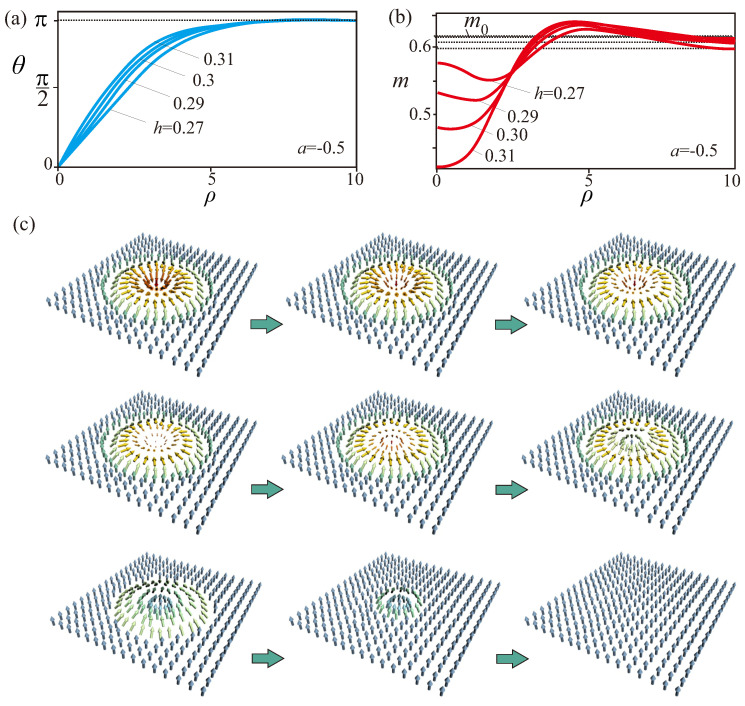
(color online) Increasing magnetic field applied to isolated skyrmion (a=−0.5) localizes profiles θ(ρ) (**a**) and leads to the disappearance of isolated skyrmions by squeezing out modulus $m_{1}$ in the center (**b**). (**c**) Snapshots showing skyrmion transformation into the homogeneous state. The modulus $m_{1}$ in the skyrmion center gradually decreases, passes through zero and appears on the other side being aligned along the field. Although one can obtain solutions for the whole process, skyrmions collapse once the modulus approaches zero value.

**Figure 3 nanomaterials-13-00891-f003:**
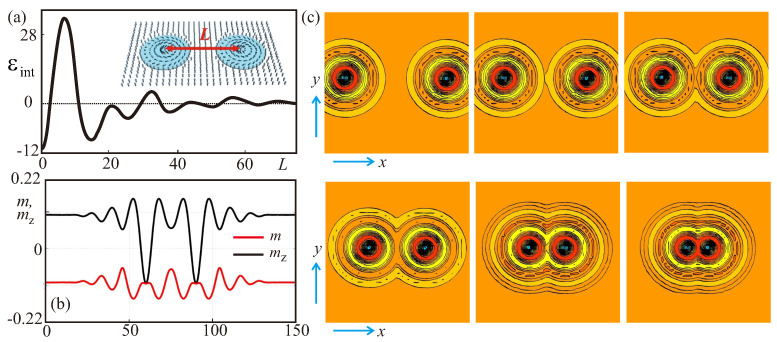
(color online) (**a**) The skyrmion–skyrmion interaction energy $\varepsilon_{\mathrm{int}}$ plotted in dependence on the distance *L* between the centers of two isolated skyrmions. The energy $\varepsilon_{\mathrm{int}}$ exhibits a number of local minima which imply attracting skyrmion interaction; (**b**) Dependencies of the modulus *m* (red line) and $m_{z}$-component of the magnetization (black line) in the cross-section of two interacting isolated skyrmions for a=0.21,h=0.048 corresponding to the first (deepest) minimum of the interaction energy $\varepsilon_{\mathrm{int}}$; (**c**) color plots of $m_{z}$-component of the magnetization showing skyrmion pair configurations corresponding to the local minima of $\varepsilon_{\mathrm{int}}$.

**Figure 4 nanomaterials-13-00891-f004:**
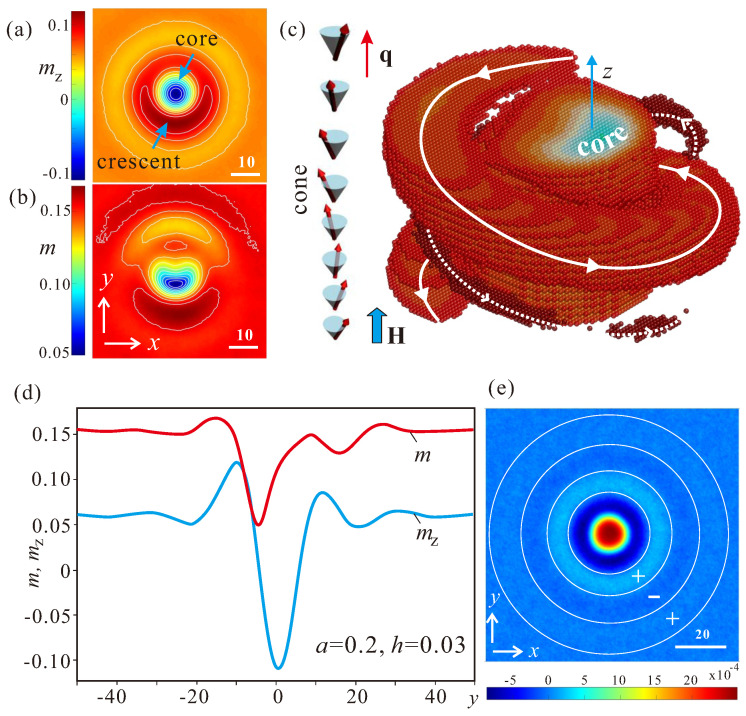
(color online) Magnetic structure of nonaxisymmetric skyrmions within the conical phase of bulk cubic helimagnets near the ordering temperature. Color plots of the out-of-plane magnetic moment, $m_{z}\left(x,y\right)$ (**a**), and the modulus, m(x,y) (**b**), plotted for one cross-section with the fixed coordinate *z*. In (**c**), an alternative way of representing the internal structure of a nonaxisymmetric skyrmion is used. After the magnetization components corresponding to the conical phase have been extracted, the 3D-model of the modulus *m* represents a cylinder-like core centered around the magnetization opposite to the field and exhibiting some reduced value and two coils with the magnetization along the field and enhanced modulus value. (**d**) two-dimensional cross-cuts across the skyrmion center exhibiting distributions $m_{z}\left(y\right)$ and m(y). After averaging over all energy profiles within one conical period, the energy distribution (**e**) exhibits the following composite parts: the core with the positive energy density, the ring with the negative energy density, and a number of additional rings with alternating energy density.

## Data Availability

The data presented in this study are available on request from the corresponding author.

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
