# Peer review of "Mechanism of Skyrmion Attraction in Chiral Magnets near the Ordering Temperatures"

_nanomaterials, 2023, doi:10.3390/nano13050891_

Round 1
Reviewer 1 Report
Authors examined isolated chiral skyrmions within the phenomenological Dzyaloshinskii model near the ordering temperatures of quasi-two-dimensional chiral magnets with Cnv symmetry and three-dimensional cubic helimagnets in this manuscript. In the first case, the attraction between particle-like states is repulsive at low temperatures but becomes attractive at high temperatures. In the latter case, they observe that the nascent conical state in bulk cubic helimagnets appears to shape skyrmion's internal structure and support their mutual attraction.
This work, according to the author, provides fundamental insights into the mechanism for complex mesophase formation close to the ordering temperatures and is a first step in explaining the phenomenon of numerous precursor effects in this temperature region.
The authors have performed extensive research using theory and modeling. Regarding the experimental demonstration, I would advise authors to review Eric's paper, which can be found at https://pubs.acs.org/doi/abs/10.1021/acsnano.9b08699, to determine if they can relate their work to this or to other experimental outcomes.
On a related point, it seems as though the quality of the modelling efforts has been improved. Nevertheless, the authors need to provide citations for any borrowed equations, if they have any.
Reviewer 2 Report
Andrey and Ulrich theoretically investigated the isolated skyrmions and their interactions at different temperatures. They obtained a phase contour that identify isolated skyrmions as the ground state at the ordering temperature, below (above) which skyrmion interaction is repulsive (attractive). I find their work interesting and meaningful for understanding the various metastable phases in chiral magnetic materials. I only have some minor comments as follows:
(1) The authors negnect several energy contributions , such as magnetic anisotropy energy and demagnetizing-field energy. As these contributions play an important role in stabilizing skyrmions in thin film systems, I am wondering if it is the same case for the bulk materials, such as MnSi.
(2) For the free energy f(m), why a1 = 0 corresponds to the critical temperature and what is the physical meaning for the coefficient a2.
(3) The authors discussed in detail the skyrmion property in the frustrated materials in the introduction part, which seems not very relevant to the main contents in this work.
